# Statistical Analysis of Maximally Similar Sets in Ecological Research

**David W. Roberts**

Montana State University, Bozeman, MT 59717, USA; droberts@montana.edu; Tel.: +1-406-570-3447

**Abstract:** Maximally similar sets (MSSs) are sets of elements that share a neighborhood in a high-dimensional space defined by a symmetric, reflexive similarity relation. Each element of the universe is employed as the kernel of a neighborhood of a given size (number of members), and elements are added to the neighborhood in order of similarity to the current members of the set until the desired neighborhood size is achieved. The set of neighborhoods is then reduced to the set of unique, maximally similar sets by eliminating all sets that are permutations of an existing set. Subsequently, the within-MSS variability of candidate explanatory variables associated with the elements is compared to random sets of the same size to estimate the probability of obtaining variability as low as was observed. Explanatory variables can be compared for effect size by the rank order of within-MSS variability and random set variability, correcting for statistical power as necessary. The analyses performed identify constraints, as opposed to determinants, in the triangular distribution of pair-wise element similarity. In the example given here, the variability in spring temperature, summer temperature, and the growing degree days of forest vegetation sample units shows the greatest constraint on forest composition of a large set of candidate environmental variables.

**Keywords:** similarity relation neighborhoods; similarity relation decomposition; statistical analysis of within-set variability

## 1. Introduction

The discipline of community ecology investigates variability in the composition of ecological communities and the possible environmental factors that might determine or constrain that composition. Because ecological communities generally contain multiple species, the comparison of composition is inherently multivariate. The general approach to the analysis of this variability involves the calculation of a symmetric matrix of similarities, dissimilarities, or distances between all possible pairs of community sample units, followed by the subsequent analysis of the properties of that matrix. This analysis typically takes one of two forms: partitioning the sample units into sets of similar samples through some form of cluster analysis, or projecting the matrix to lower dimensionality and analyzing the variability as a field, which ecologists refer to as "ordination." Textbook treatments of these methods are given by Legendre and Legendre [1] and Kent [2].

Formal statistical analysis of ecological communities goes back to at least 1954, when David Goodall introduced the use of principal component analysis (PCA) to ecology [3]. Over the subsequent decades, numerous statistical methods have been adopted or invented to further the aims of community ecology. In particular, various forms of cluster analysis have been employed in ecological analysis. Recent examples include analyses of plants [4–6], fish [7], birds [8,9], and mammals [10]. Examples of comparative analyses of clustering algorithms applied to ecological community data are given by Roberts [11] and Aho et al. [12]. Generally, if not without exception, ecological cluster analyses have produced partitions of the sample units, i.e., a family of sets where (1) every set has at least one member, (2) every element belongs to exactly one set, and (3) the union of sets comprises the universe

of elements. While there are practical reasons to desire a partition as the outcome of the analysis (e.g., to produce a set of community types for inventory or mapping purposes), the use of partitions presents challenges in statistical analysis of the underlying environmental constraints or determinants of the community type composition.

Given a sample with $N$ sample units, the number of possible partitions of the data into clusters is given by Bell's number

$$B_N = \sum_{k=0}^{N} \left\{ \begin{matrix} N \\ K \end{matrix} \right\}. \tag{1}$$

where $K$ is a given number of clusters ($0 \leq K \leq N$), $N$ is the number of sample units, and $\left\{ \begin{smallmatrix} N \\ K \end{smallmatrix} \right\}$ is the Stirling number of the second kind.

$$\left\{ \begin{matrix} N \\ K \end{matrix} \right\} = \frac{1}{K!} \sum_{j=0}^{K} (-1)^{K-j} \binom{K}{j} j^N. \tag{2}$$

While the population of possible partitions is generally large, sampling at random from that population is difficult due to the peculiarities of the cluster analysis result to be compared to. Even constraining the sample population to the same value of $K$, it may or may not be desirable to constrain the individual clusters to the distribution of cluster sizes obtained in the original cluster analysis, which is in part an artifact of the particular cluster analysis performed and not necessarily a good result.

Alternatively, it is possible to calculate a covering, as opposed to a partition of the data, i.e., a family of sets such that (1) all sets have at least one member, (2) all elements belong to at least one set, and (3) the union of all sets comprises the universe of sample units. By relaxing the partition constraint that every element belongs to exactly one set, but imposing the constraint that all sets are the same size, we can derive a covering, as opposed to a partition. In this case, the number of possible sets of size $n_k$ in the covering for $K$ clusters is

$$\binom{N}{n_k} \tag{3}$$

where $n_k$ is the number of elements of set $k \in K$. This is a much more favorable population to sample from. We simply sample $n_k$ sample units without replacement from the set of $N$ sample units a large number of times and compare the random samples to the the observed sets in the covering.

The specific objectives of this paper are to demonstrate and test the utility of the proposed analysis on a specific ecological data set. The primary hypothesis underlying the test is that ecological community composition is constrained (as opposed to determined) by environmental factors acting upon the individual species that make up the community, and that such factors can be identified by a test designed to identify limiting conditions within sets of similar communities. The concept of maximally similar sets and the specific algorithms for construction and analysis of such sets by permutation employing non-parametric statistics are novel and have not previously been demonstrated in the scientific literature.

## 2. Materials and Methods

The proposed analysis is demonstrated on a sample of the forest vegetation of the Shoshone National Forest, Wyoming, USA. The vegetation composition varies primarily as a function of environmental variability, but the precise nature of this relationship is unknown. Maximally similar sets analysis as described above is employed to tease out this relationship.

### 2.1. Data

Sample units were 375 m$^2$ circular plots, where the abundance of every vascular plant species was estimated according to an ordinal ten-class scale. Environmental attributes associated with the sample

units were either measured in the field or modeled in a geographic information system. Attributes include sample unit elevation, aspect, slope steepness, surficial geology, soil properties, topographic position, and climate attributes modeled from elevation, aspect, slope, surficial geology, topographic position, and geographic location. One hundred fifty sample units were selected at random from a larger study to provide a manageable example data set. Further details about the data are given in Appendix A.

*2.2. Analyses*

The sample unit composition data were used to calculate a symmetric, reflexive similarity matrix using the Bray–Curtis index [13].

$$s_{ij} = \frac{\sum_{q=1}^{m} 2 \times min(a'_{iq}, a'_{jq})}{\sum_{q=1}^{m} a'_{iq} + a'_{jq}} \tag{4}$$

where $s_{ij}$ is the similarity of sample unit $i$ to sample unit $j$, $m$ is the number of species, $a_{iq}$ is the abundance of species $q$ in sample unit $i$, and $a'_{iq} = log(a_{iq} + 1)$. $s_{ij} \to [0, 1]$ where sample units with no species in common $= 0$ and identical sample units $= 1$.

Maximally similar sets (MSSs) were solved for by setting the desired neighborhood size $n_k$ and then, iteratively for each sample unit, adding the sample unit most similar to the members of the neighborhood until the desired neighborhood size was achieved. The most similar sample unit to the neighborhood was calculated as

$$s_{ik_x} = \max_{\substack{i=1 \\ i \ni k_x; j \in k_x}}^{N} \min s_{ij} \tag{5}$$

where $s_{ik_x}$ is the similarity of sample unit $i$ to neighborhood $k_x$.

The set of resulting neighborhoods was reduced to the set of unique neighborhoods by deleting all neighborhoods that were a permutation of an existing neighborhood. The number of neighborhoods in the similarity relation ($S$) reflects the topology of the similarity relation and cannot be known *a priori*. As the size of neighborhoods increases, the number of neighborhoods generally declines.

The within-MSS variability of interval- or ratio-scaled sample unit attributes was determined by calculating the range of the attribute within each MSS. That set of $K$ ranges was then compared to the set of ranges of an equal number of sets of equal size drawn at random without replacement from the set of sample units. The ranges of the MSS were compared to the ranges of the randomly drawn sets in a Wilcoxon rank-sum test with continuity correction to generate the Wilcoxon statistic $W$. The within-MSS variability for categorical variables was determined by calculating the entropy of the tabulated values of the variable for each of the $K$ MSSs and comparing it with the entropy of $K$ sets of the same size sampled at random without replacement from the set of sample units. The entropy for set $k$ was calculated as

$$e_k = - \sum_{\substack{c=1 \\ pc \neq 0}}^{C} p_c \times \log(p_c); \quad p_c = n_c / N \tag{6}$$

where $n_c$ is the number of sample units in category $c$ for categorical variable $C$. As for the interval-valued attributes, the observed and random entropies were then tested with a Wilcoxon rank sum test with continuity correction to generate the Wilcoxon statistic $W$. This process was repeated 1000 times for each attribute to generate 1000 $W$ statistics for each attribute, and the effect size of each attribute was estimated by comparing the distribution of $W$ values in a boxplot.

## 3. Results

The analysis generated neighborhoods of sample units of size 5, 10, 15, and 20, where each neighborhood consisted of sample units with maximally similar composition. Distributions of Wilcoxon's *W* were produced for each candidate explanatory variable for all neighborhoods of all sizes. The number of neighborhoods generated and the effect size analysis of the variables is presented below.

### 3.1. Similarity Relation and Maximally Similar Sets

The distribution of similarities in the Bray–Curtis similarity relation is shown in Figure 1. Figure 1a shows relatively few cases of $s_{ij} = 0$, and relatively few cases of $s_{ij} > 0.4$. Figure 1b shows that the mean similarity and minimum similarity are somewhat correlated, and that most sample units have at least one other sample unit with $s_{ij} > 0.5$. This is a typical result for vegetation data where the composition of the sample units is quite diverse but where sampling is adequate to span the range of variability.

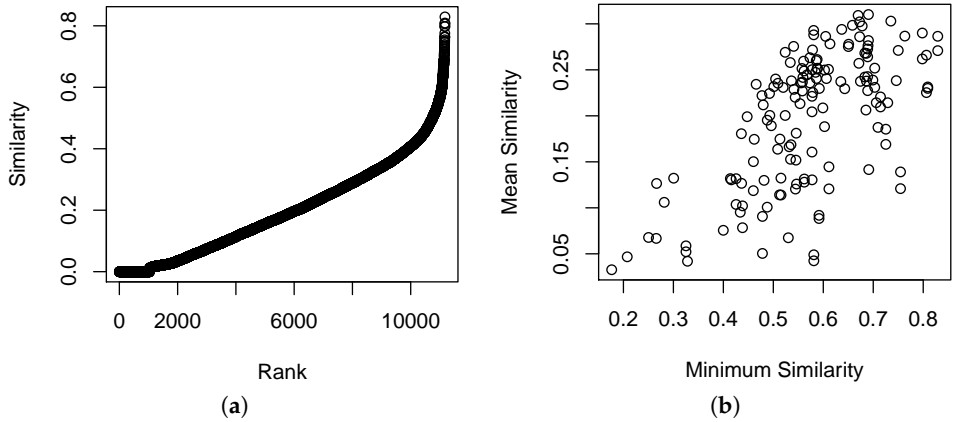

(a)                    (b)

**Figure 1.** Distribution of similarity values in the Bray–Curtis similarity relation. (**a**) The cumulative empirical density distribution. (**b**) The pairwise distribution of minimum and mean similarity for each sample unit.

MSSs were calculated for neighborhood sizes of 5, 10, 15, and 20 members. Table 1 gives the sizes of the resulting families of neighborhoods.

**Table 1.** Number of neighborhoods as a function of neighborhood size.

| Neighborhood Size | Number of Neighborhoods |
| --- | --- |
| 5 | 102 |
| 10 | 90 |
| 15 | 78 |
| 20 | 71 |

Setting $n_k = 10$ resulted in 90 MSSs, and this result is used in subsequent analyses. Given $N = 150, K = 90$, and $n_k = 10$, on average, sample units belonged to six neighborhoods. In many cases, the similarity of the second element joining the set is low, but the monotonic decline in the similarity of subsequent elements is relatively gentle (Figure 2); the lowest similarity of any member of a set ranges from 0.067 to 0.50 with a median of 0.35.

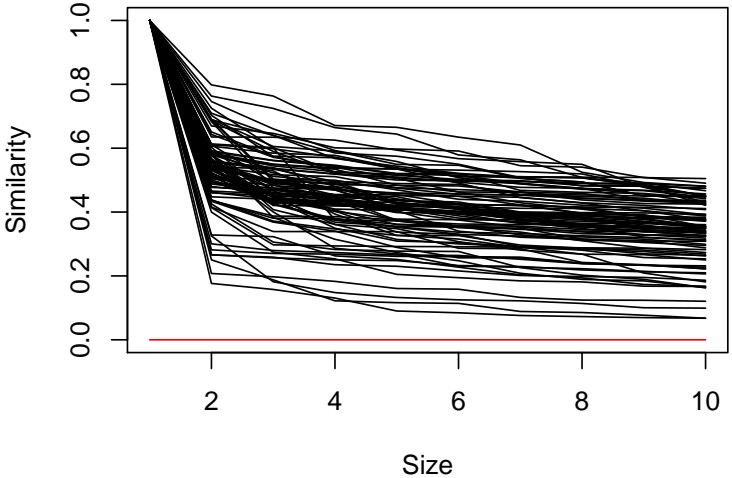

**Figure 2.** Trace of similarity values for each of the 90 MSSs as new elements were added to the sets.

### 3.2. Analysis of Environmental Constraints

Figure 3 shows the distribution of Wilcoxon's *W* for a broad range of interval- or ratio-scaled environmental attributes, mostly related to topography and climate. Boxes A–C represent commonly observed plot-level observations: elevation (m), aspect value $((\cos(aspect - 30°) + 1)/2)$, and slope steepness in percent. Elevation shows quite a strong constraint, where sample units with similar vegetation composition must occur within a relatively narrow band of elevation; aspect and slope show very little constraint. Boxes D–G are seasonal precipitation and show moderate constraint on similarity. Boxes H–K show seasonal temperature and, except for winter temperature (H), show quite a significant constraint; spring temperature has the strongest constraint of any attribute, and summer temperature ranks third in effect size. Boxes L–O show direct solar radiation (heat load) and exhibit little direct constraint. Boxes P–R show seasonal potential evapotranspiration (PET); spring PET ranks fourth in effect size, and summer and autumn PET rank ninth and tenth, respectively. Boxes T–V show annual climatic summaries: the mean annual temperature, the number of frost free days, and the growing degree days (sum of temperature $> 5\,°C$). All three variables show strong constraint; growing degree days ranks second in effect size, the number of frost free days ranks fifth, and the mean annual temperature ranks seventh.

Figure 4 shows the the distribution of Wilcoxon's *W* for a range of categorical variables, mostly related to soil properties and geology. Boxes A–C show the constraints associated with typically observed soil properties. Of the three, soil depth (classified into four categories) shows the greatest constraint. Boxes D–F show the effect size of soil classifications commonly employed in the United States. Soil short family (D) was classified into 58 classes with many singletons and one class with 21 sample units. Soil subgroup was classified into 16 classes with a few singletons and one class with 50 sample units. Soil great group (F) showed the greatest constraint of the three and was classified into seven classes with two singletons and the maximum of 63 sample units/class.

Surficial geology was classified into 23 classes, with several singletons and a maximum or 30 sample units/class, and showed the greatest constraint of any of the categorical attributes. Sample units with similar vegetation are likely to occur on a narrow range of surficial geology types. In general, among the interval-scaled attributes, the greatest constraint was demonstrated by variables associated with sample unit temperatures, either directly or indirectly. Spring potential evapotranspiration also ranked in the top five but is also a function of temperature. Surprisingly, precipitation generally showed little effect, with seasonal precipitation ranking 11–14. Solar radiation showed very little effect, suggesting that the same level of radiation can occur in sites of quite different vegetation composition if the temperature of the units is different. Among the categorical variables only surficial

geology showed a substantial constraint, exhibiting values similar to seasonal precipitation among the interval-scale variables.

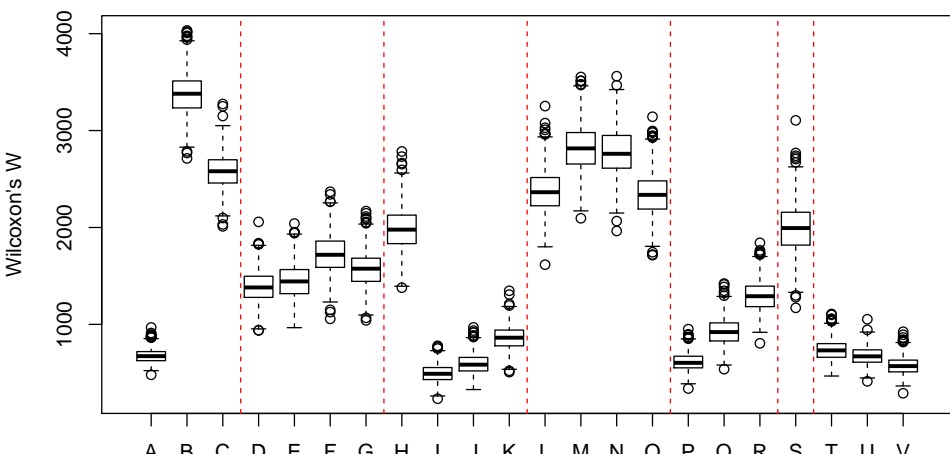

**Figure 3.** Distribution of Wilcoxon's *W* for selected environmental attributes with neighborhood size set to 10. A = elevation (m); B = aspect value; C = slope steepness; D = winter precipitation (mm); E = spring precipitation (mm); F = summer precipitation (mm); G = autumn precipitation (mm); H = winter temperature (°C); I = spring temperature (°C); J = summer temperature (°C); K = autumn temperature (°C); L = winter radiation (W/m$^2$); M = spring radiation (W/m$^2$); N = summer radiation (W/m$^2$); O = autumn radiation (W/m$^2$); P = spring potential evapotranspiration (mm); Q = summer potential evapotranspiration (mm); R = autumn potential evapotranspiration; S = site water balance (precipitation - potential evapotranspiration); T = mean annual temperature (°C); U = frost free days; V = growing degree days. Red dashed lines separate logically related attributes.

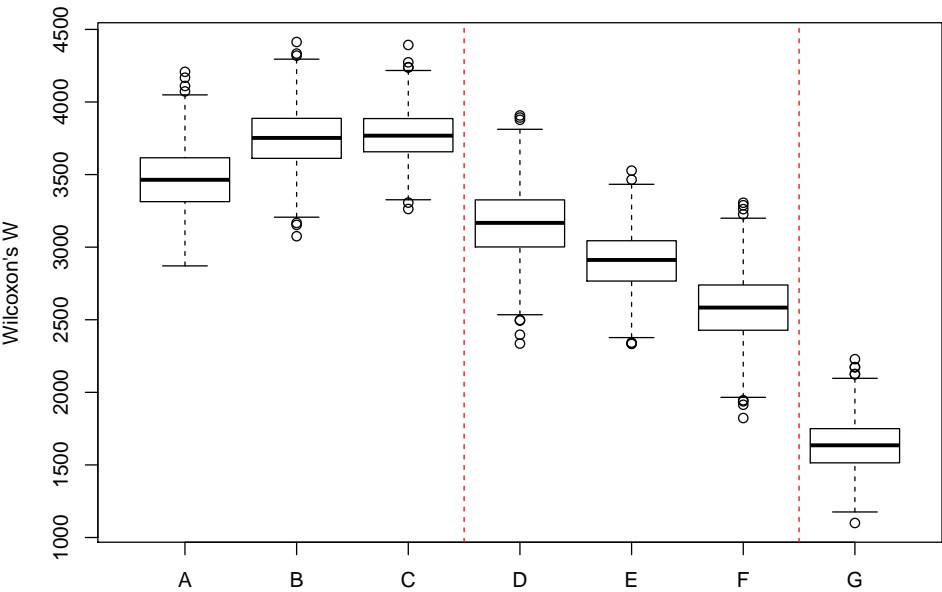

**Figure 4.** Distribution of Wilcoxon's *W* for selected environmental attributes with neighborhood size set to 10. A = soil depth; B = soil texture; C = coarse *vs* fine texture; D = soil short family; E = soil subgroup; F = soil great group; G = surficial geology. Red dashed lines separate logically related attributes.

## 4. Discussion

In general, the method worked well at identifying primary ecological constraints of community composition in the data examined. However, there are several methodological considerations to consider in applying the method to a given data set.

### 4.1. Constraint vs. Determinant

Throughout the results, I have repeatedly specified "constraint" as opposed to "determinant" in interpreting the results. The composition of ecological communities (forest vegetation in this case) is determined by a complex set of processes. While species have individual responses to specific environmental attributes, the form of integration of those individual responses into the overall species response is generally unknown. Often, environmental attributes can be partly compensatory, and the relationships are generally not linear or independent. Notably, a suitable habitat is necessary but not sufficient, so that species may be absent from a community for reasons unrelated to the environment at that location. Accordingly, the relationship between environment and community composition is a relation, as opposed to a function.

Figure 5 shows the relationship between the pair-wise compositional similarities and the pair-wise differences in sample unit elevation. The plot shows the classical triangular distribution characteristic of these relationships. As the difference in elevation between sample units increases, it becomes increasingly difficult to compensate for that difference, and maximum possible sample unit similarity declines. However, even at zero or small differences in elevation, compositional similarity can be zero or low. Given the triangular distribution of the similarity/environment relation, the MSS analysis proposed here looks for boundary conditions, i.e., "how different can values of an environmental attribute be while still allowing similar community composition?" While it may appear that such relations are suitable for analysis by logistic quantile regression [14], it is important to note that the points on the figure are not independent of each other and that in fact every point is associated with $N - 1$ other points because they pertain to the same sample unit.

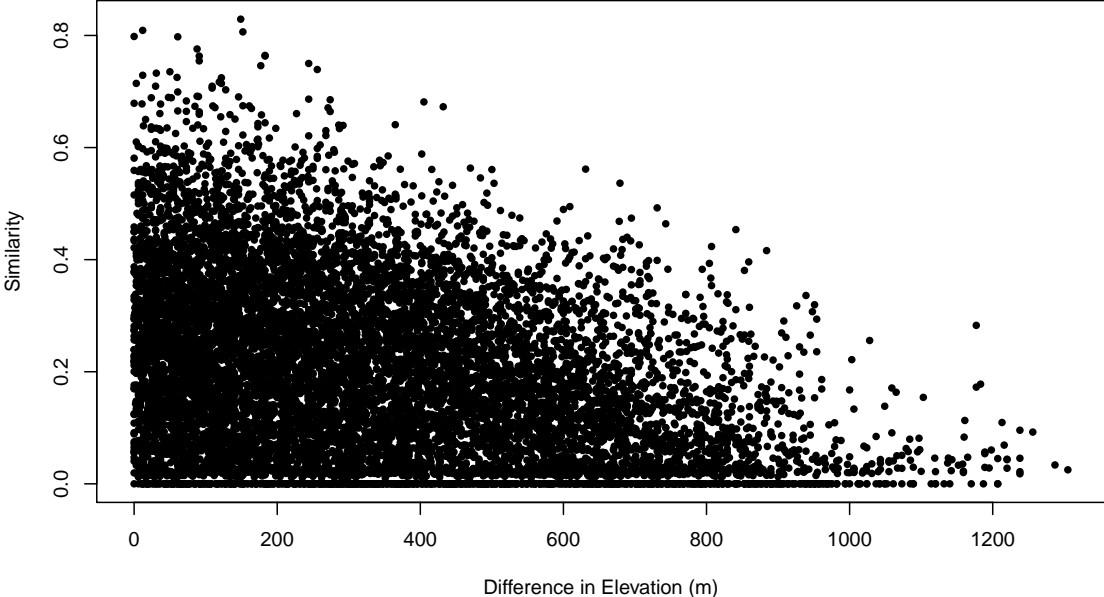

**Figure 5.** Distribution of sample unit pair-wise similarity as a function of sample unit pair-wise difference in elevation (m).

### 4.2. Neighborhood Size

The calculation of maximally similar sets was conducted at sizes of 5, 10, 15, and 20. Specific results were only shown for sets of size 10. Across the range of neighborhood sizes examined, the results were quite similar (Spearman rank correlations for the effect size of the candidate explanatory variables were in the range $[0.9282656, 0.9910036]$). In a sense, this is quite reassuring and demonstrates low sensitivity of the analysis to this parameter. Nonetheless, neighborhood size does matter in the analysis. The algorithm looks for boundary conditions within the neighborhoods. The number of pair-wise differences in a neighborhood (and thus statistical power) scales as $(n_k^2 - n_k)/2$. Accordingly, small neighborhoods may by chance not exhibit any sample unit pairs at the limit for a given environmental attribute, and thus larger neighborhoods are preferred. However, as neighborhood size increases, the number of neighborhoods generally declines, and the power of the test thus declines as well, although modestly. In the data analyzed here, the number of neighborhoods declined from 102 to 71, as neighborhood size increased from 5 to 20. In addition, as neighborhood size increases, the similarity to the neighborhood of late-joining sample units declines, so quite large pair-wise differences in environment may be obtained. At the extreme, if the neighborhoods are too large, sample plots will be added to neighborhoods that have no similarity to other members of the neighborhood and thus contribute no information. The optimal neighborhood size is thus a function of the size of the data set and the distribution of similarities in the similarity relation, and generally cannot be known *a priori*. Figure 6 shows the distribution of element-to-neighborhood similarities as a function of neighborhood size for the data analyzed.

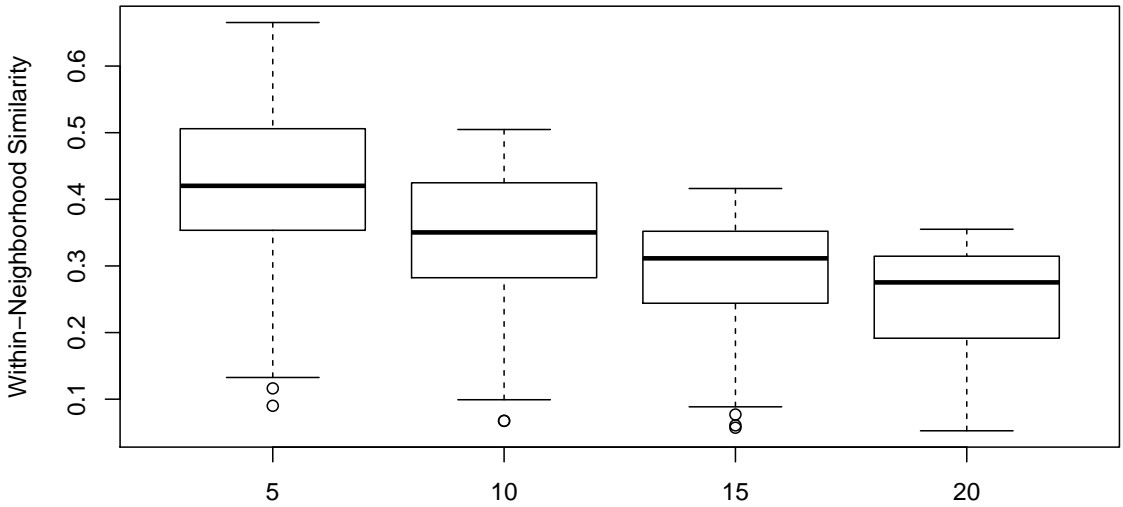

**Figure 6.** Distribution of sample unit pair-wise similarity within neighborhoods as a function of neighborhood size.

### 4.3. Interval-Scaled vs. Categorical Variables

Both interval-valued and categorical variables were analyzed using the Wilcoxon rank sum test, and thus present results on the same scale. However, depending on the number of classes within a categorical variable, it is common to be observed or random sets that have identical entropies and thus generate ties in the calculation of Wilcoxon's *W*. As the number of ties goes up, the power of the test declines, and some categorical variables thus have low power. In addition, ideally the distribution of cases by class would be relatively balanced. In the data analyzed here, the distributions were wildly skewed, which again reduces power. Of the categorical variables analyzed, only surficial geology demonstrated a strong constraint. The extent to which the other variables suffered from low power,

as opposed to a minimal ecological effect, is difficult to know, but the results obtained do make sense from an ecological perspective.

*4.4. Ecological Interpretation*

The point of any ecological analysis is to generate insight into the ecological processes and relationships at work. In this case, the signal was clear that the seasonal air temperature at the sample unit locations was the primary constraint on community similarity. Forest plant communities that are similar to each other must occur within a narrow range of temperatures. This makes sense in that plant ecophysiological behavior is a function of temperature. At low temperatures, plants photosynthesize less efficiently and thus gain less carbon for growth or maintenance. At high temperatures, plants suffer from excess sensible heat, and net photosynthesis again declines. Each plant species has an optimal temperature, and plants with similar optima are thus likely to co-occur in sample units with that temperature. Interestingly, the highest ranked variable was specifically the spring temperature (April, May, and June), which may indicate critical timing of the beginning of growth after dormancy in the winter. Summer temperature ranks third in effect size, so temperature throughout the growing season seems quite important. The second strongest constraint is growing degree days, which is the sum of temperatures above 5 °C, and again strongly determined by temperature.

Interestingly, solar radiation showed little effect. Solar radiation contributes heat to the sites of the sample units that can be partitioned into latent heat of vaporization (evapotranspiration of water) or sensible heat. Apparently, sites receiving similar radiation budgets but differing in available water differ in the partitioning of heat into latent heat and sensible heat and can thus compensate for differences in radiation.

Surprisingly, precipitation showed relatively little effect, with winter (January, February, and March) precipitation (which comes primarily as snow) showing the largest effect, ranked only at 11th. Water is a necessary resource for plant metabolism and plays a large role in the partitioning of heat into latent heat. However, precipitation does not translate directly into plant-available water and interacts with temperature-controlled evapotranspiration. It is possible to model potential evapotranspiration (the amount of water the atmosphere could evaporate if water was not limiting). In the data analyzed here, the potential evapotranspiration of summer (July, August, and September) and autumn (October, November, and December) ranked eighth and ninth, respectively.

Variability in soils plays a critical role in plant species distribution and community composition, but is notoriously difficult to code as an explanatory variable in statistical analyses. Of the soils variables considered here, surficial geology played the largest role. Surficial geology determines what soil scientists call the soil "parent material" and affects soil water-holding capacity and fertility. Soil parent materials are complex multi-faceted variables that have to be treated as categorical variables in an analysis like this one, which ignores the similarities or differences among types. Nonetheless, soil parent material makes sense as a community composition constraint, as many species are known to have preferences for specific types.

## 5. Conclusions

Analysis of maximally similar sets operates on a similarity matrix to define neighborhoods in the high-dimensional similarity space defined by the matrix. After forming the neighborhoods, the analysis evaluates within-neighborhood variability in candidate explanatory variables to obtain an estimate of variable effect size. The probability of obtaining results with variability as low as observed is calculated by sampling a large number of sets the same size as the neighborhood and calculating a Wilcoxon siged-rank test of the results. The analysis of maximally similar sets proved to be an effective analysis of variables that exhibit a relational, as opposed to functional, response. The analysis effectively ranked a set of candidate explanatory variables to provide an interpretable ecological analysis. The algorithm is reasonably quick on data sets of this size (the neighorhood analysis with a neighborhood size set at 10 required 0.03 s to run) but would take longer on significantly larger data sets. The analysis provides

a more suitable statistical population for sampling than do partition-based cluster analyses, and avoids the artifact characteristic of hierarchical cluster analyses.

The analysis on the example data set of forest vegetation identified sample unit temperature (in multiple forms) as the primary constraint on neighborhood composition, followed by potential evapotranspiration. While the analysis presented here concerned forest vegetation, the analysis is suitable for any multivariate data set, where the relationship among sample units can be described with a similarity matrix, such as that employed in ecology, psychology, and machine learning fields.

**Funding:** This research received no external funding

**Conflicts of Interest:** The author declares no conflict of interest

## Abbreviations

The following abbreviations are used in this manuscript:

MSS     maximally similar set
PET     potential evapotranspiration

## Appendix A

The Shoshone National Forest in Wyoming, USA, is part of the United States Department of Agriculture National Forest System. In a previous analysis of plant community composition, 1204 sample units of 375 m$^2$ were stratified throughout the National Forest. The total abundance of all individuals of each vascular plant species was estimated as the percent of the plot area covered by foliage of that species recorded in the classes given by Table A1.

**Table A1.** Species abundance classes.

| Range | Mid-Point Abundance |
|-------|---------------------|
| $0 \leq 1$ | 0.5 |
| $1 \leq 5$ | 3 |
| $5 \leq 15$ | 10 |
| $15 \leq 25$ | 20 |
| $25 \leq 35$ | 30 |
| $35 \leq 45$ | 40 |
| $45 \leq 55$ | 50 |
| $55 \leq 65$ | 60 |
| $65 \leq 75$ | 70 |
| $75 \leq 85$ | 80 |

Of the original 1204 sample units, 150 sample units of forest communities were drawn at random from the pool for the analysis presented here. The natural log of the mid-point abundances given in Table A1 were used in the calculation of the sample unit similarity relation as given in Equation (4).

Environmental variables were either recorded in the field or modeled in a GIS. Field measured variables are in Table A2.

**Table A2.** Field Measured Variables.

| Variable | Description |
|---|---|
| elevation | elevation in meters above sea level |
| aspect value | $(\cos(aspect - 30) + 1)/2$ |
| slope | slope gradient in percent |
| soil depth | shallow, moderate, deep, or very deep |
| parent material class | coarse loamy, fine loamy, loamy skeletal, sandy skeletal, or fragmented |
| texture | coarse or fine |
| short family | USDA Soils classification (58 classes) |
| soil subgroup | USDA Soils classification (16 classes) |
| soil great group | USDA Soils classification (7 classes) |
| surficial geology | geologic rock type (23 classes) |

GIS-modeled data are in Table A3.

**Table A3.** GIS-modeled variables.

| Variable | Description |
|---|---|
| Winter | January, February, and March<br>    sum of precipitation (mm)<br>    mean monthly temperature (°C)<br>    solar radiation (W/m$^2$) |
| Spring | April, May and June<br>    sum of precipitation (cm)<br>    mean monthly temperature (°C)<br>    solar radiation (W/m$^2$)<br>    potential evapotranspiration |
| Summer | July, August, September<br>    sum of precipitation (cm)<br>    mean monthly temperature (°C)<br>    solar radiation (W/m$^2$)<br>    potential evapotranspiration |
| Autumn | October, November, and December<br>    sum of precipitation (cm)<br>    mean monthly temperature (°C)<br>    solar radiation (W/m$^2$)<br>    potential evapotranspiration |
| Frost-free days<br>degree days<br>site water balance<br>mean annual temperature | length of growing season in days<br>sum of hours $> 5\,°C$<br>sum of (precipitation - potential evapotranspiration)<br>°C |

The sample unit vegetation and site data are available at Figshare at https://doi.org/10.6084/m9.figshare.7234121.v1.

The R code for the analysis is available at Figshare at https://doi.org/10.6084/m9.figshare.7234124.v1.

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
