# Peer review of "Statistical Analysis of Maximally Similar Sets in Ecological Research"

_mathematics, doi:10.3390/math6120317_

Reviewer 1 Report

General impression:

The paper provides a good insight into typical problem of ecological research: How can  cological communities be determined from the floristic composition of plots? In this article, the author  hows the value of using overlay rather than data partition to highlight ecological communities.

The manuscript should be published after some modifications. The author has to add a iscussion and improve conclusions of the research.

Introduction:

The introduction would become clearer by adding references: e.g. l. 24-26 references would be appreciated. Similarly l. 30, referenced examples would convince the reader.

Materials and Methods:

The accuracy of the data set and the methods used is appreciated.

Eq. (5): i does not belong to k_x ?

The author talks about neighbourhood without the notion being well defined. Neighbourhood in the sense of ecological communities? Besides in the paper, there is no analysis on the ecological communities that emerge. An analysis would be appreciated.

This notion of neighbourhood and covering using maximally similar set is fundamental for the future because the variables are explained from these sets. Details would therefore be  ppreciated on the composition of these sets, to explain (perhaps?) the order of the variables selected.

Results:

3.2 Analysis of Environmental Constraints

l. 142-144: Can you explain why solar radiation is linked to the average temperature ot units ? Do you mean emissivity temperature ?

Discussion:

4.2. Neighborhood size

The explanation of the optimal size of the neighbourhood is not clear. The author explains well that he proceeded in steps of 5 between 5 and 20, but the impact of size is not quantitatively

explained.

A conclusion would be appreciated. Are there any possibilities for other types of environment? fields ? grasslands ? wetlands?

Reviewer 2 Report

Article is well postioned in journal's scope and is written according to rules.

I suggest extend bbibligraphy list using relevant and up to date studies.

Additionally "Conclusions" or "Summary" section should be provided

Reviewer 3 Report

The author(s) of the manuscript submitted suggest a work about statistical analysis of maximally similar sets in ecological research. The manuscript is clearly written and easy to follow, and it is within the journal scope.

I would like to suggest the author(s) to put more extensive introduction for the general journal readers. At the same context, it would be also helpful to a small summary paragraph in the end of introduction section.

Please give the objectives and hypotheses of this study in the end of introduction.

Please insert a small paragraph between Section 2 and Section 2.1., between Section 3 and Section 3.1., and between Section 4 and Section 4.1.

Conclusions should be included in a more concisely way and compared with similar studies. The discussion should be improved. There needs to be more comparative analysis with other studies.

Figures are not clear enough so they are necessary to increase their quality and size for the readability for the general journal readers.

References are not enough and outdated. Please include the references and also update more recent researches.

References in the text and references list are particularly not followed by the journal guidelines and please include DOI if they have.

Author Response

see attached

Round  2

Reviewer 2 Report

There is still lack of extended reference list and conclusion section in provided pdf manuscript.

Author Response

Dear Reviewer,

   Please accept my sincere apology.  The manuscript was revised specifically to respond to your comments, but somehow the original PDF version, as opposed to the revised PDF version, was included in the zip file.  The updated LaTeX file was included, but the revised PDF was not.  I have corrected that.  Please note that (1) an updated list of references is provided in the introduction, and (2) a Conclusion section is included.  Again I apologize for wasting your time due to my error.

Sincerely,  Dave Roberts

Reviewer 3 Report

First of all, I appreciate to the authors for making efforts to carry out the changes by the referees. I think the authors did a good job in clarifying the queries that this manuscript is substantially improved. 

Author Response

Dear Reviewer,

    Thank you for your comments.

Sincerely, Dave Roberts